# Feature-Selection-Based Attentional-Deconvolution Detector for German Traffic Sign Detection Benchmark

Junho Chung [1], Sangkyoo Park [1], Dongsung Pae [2,*], Hyunduck Choi [3] and Myotaeg Lim [1,*]

1 School of Electrical Engineering, Korea University, Seoul 02841, Republic of Korea
2 Department of Software, Sangmyung University, Cheonan 31066, Republic of Korea
3 Department of ICT Convergence System Engineering, Chonnam University, Gwangju 61186, Republic of Korea
* Correspondence: paeds915@smu.ac.kr (D.P.); mlim@korea.ac.kr (M.L.)

**Abstract:** In this study, we propose a novel traffic sign detection algorithm based on the deep-learning approach. The proposed algorithm, which we termed the feature-selection-based attentional-deconvolution detector (FSADD), is used along with the "you look only once" (YOLO) v5 structure for feature selection. When applying feature selection inside a detection algorithm, the network divides the extracted feature maps after the convolution layer into similar and non similar feature maps. Generally, the feature maps obtained after the convolution layers are the outputs of filters with random weights. Owing to the randomness of the filter, the network obtains various kinds of feature maps with unnecessary components, which degrades the detection performance. However, grouping feature maps with high similarities can increase the relativeness of each feature map, thereby improving the network detection of specific targets from images. Furthermore, the proposed FSADD model has modified sizes of the receptive fields for improved traffic sign detection performance. Many of the available general detection algorithms are unsuitable for the German traffic sign detection benchmark (GTSDB) because of the small sizes of these signs in the images. Experimental comparisons were performed with respect to the GTSDB to show that the proposed FSADD is comparable to the state-of-the-art while detecting 29 kinds of traffic signs with 73.9% accuracy of classification performances.

**Keywords:** traffic sign detection; feature selection; YOLO; German traffic sign detection benchmark

## 1. Introduction

Many studies on autonomous driver assistant systems have been proposed in recent times, owing to improvements in traffic sign recognition [1–3]. Various computer vision algorithms have been applied to traffic sign recognition in the past; however, deep-learning-based algorithms [4] have attracted attention from researchers owing to their improved computing power. Nowadays, the traditional computer vision algorithms of the past have been rapidly replaced by deep-learning-based approaches because of the significant performance enhancements. Among the deep-learning-based algorithms studied widely, the convolutional neural network (CNN) has demonstrated its superior performance to other methods for traffic sign recognition [5–8].

Research on traffic sign recognition can be considered under two major areas, namely traffic sign detection and classification [9]. The main idea for traffic sign detection (TSD) involves locating traffic signs in images by drawing rectangular bounding boxes. Traffic sign classification (TSC) involves the identification of the sub classes of traffic signs within datasets.

The general principle of a detection algorithm is to find the locations of traffic signs from images or videos. Once the detection algorithms predict the possible candidate regions of the traffic signs, rectangular boxes are drawn to locate signs. Traditionally, traffic

signs have been detected using the histogram of oriented gradient features [10], the Kalman filter [11], and support vector machines [12]. However, traditional computer vision algorithms are vulnerable to external factors, such as illumination, color, blur, or occlusions, which cause difficulties in identifying the traffic signs [6]. In contrast to the traditional detection algorithms, deep-learning-based approaches have demonstrated superior performances with higher accuracies and robustness against the external factors. Therefore, deep-learning-based algorithms [13–20] have become one of the major strategies for TSD.

TSC has become one of the major research areas, and outstanding performances have been achieved by adopting CNN-based architectures. The convolution layers inside the algorithms extract feature information using specific numbers of filters with different weights. Owing to this characteristic, the high classification performance means that the algorithm obtains appropriate feature information with very small losses. Therefore, many TSD algorithms have adopted a classify layer to identify what the rectangular boxes capture in the images.

By combining TSD with TSC, detection algorithms have been improved significantly to achieve high performances on traffic-sign-related datasets. One of the common datasets used for traffic sign detection is the German traffic sign detection benchmark (GTSDB) [21–23], which contains 900 images of roads with 43 categories of traffic signs. The GTSDB dataset is one of the attractive datasets for many researchers because of its relatively small number of images and variety of traffic signs [24–26]. However, the small number of images of the traffic sign is considered as a critical issue that must be augmented for algorithms intended for real-world applications.

Therefore, various other types of datasets, such as the Tsinghua-Tencent 100K dataset [27], Swedish traffic signs dataset [28], and LiSA traffic sign dataset [29], have been reported.

With the improvements in the datasets containing traffic signs, two types of CNN-based detection algorithms have been proposed, namely two-stage and one-stage detectors. Two-stage detectors mainly adopt a single process, which create numerous candidate regions for finding the target objects. The selective search is one such approach, which draws a large number of bounding boxes in the image to predict the candidate regions of the target objects. By combining traditional heuristic approaches with the CNN, the R-CNN [30] was proposed, which used 2k region proposals with the selective search. However, drawing many bounding boxes from the selective search incurs a heavy computational burden during training. Therefore, researchers have focused on designing lighter networks while maintaining the detection performances. Thus, the Fast R-CNN [31] and Faster R-CNN [32] were proposed. The biggest difference between Fast R-CNN and Faster R-CNN is the existence of the region proposal network (RPN) for predicting the candidate regions [33]. The network structure with the RPN and CNN reduces the computational cost compared with that of Fast R-CNN, but it is still relatively inconvenient to use in real-world applications. The one-stage detector was designed to overcome the obvious defects of the two-stage detector, which demonstrated a superior performance for object detection but had an extremely slow operation time. The operation of the one-stage detector involves extracting feature information from images and provides it to the classifier directly [34]. In general, a 1-by-1 convolution layer is applied to the classifier, where the information of the bounding boxes and candidate scores of objects are stored. Since the proposal of the one-stage detector algorithm "you look only once" (YOLO) [35], fusing different sizes of images inside the feature extractor has become a key factor for performance improvement. The single shot detector [36] adopts seven different sizes of receptive fields for considering various object sizes in the images. The original YOLO's upgraded versions, namely YOLO9000 [37] and YOLOv3 [38], have demonstrated incremental detection accuracies with various datasets. In particular, the receptive fields from YOLOv3 contain three different object sizes, namely large, medium, and small, which guarantee a high performance with reduced operation times.

Currently, a new and improved version, called YOLOv7 [39], has been proposed for real-time applications. From YOLO to YOLOv7, the research trend regarding one-stage detectors has mainly targeted two considerations: achieving a high detection rate accurately and reducing the computational cost for real-time applications.

This paper presents a novel algorithm, called the feature-selection-based attentional-deconvolution detector (FSADD), for traffic sign detection, especially for the GTSDB dataset. It is known that one-stage algorithms show a strong point on the operating speed with low accuracy, while two-stage algorithms provide high accuracy with low speed. The purpose of the FSADD is to improve the performance of traffic sign recognition by combining the advantages of one- and two-stage detection algorithms.

The FSADD, which is a modified version of YOLOv5 [40], adopts the feature selection using the L1-norm approach in the backbone to group similar images [41–43]. From the convolution layer, it generates multiple feature maps followed by the number of channels; however, some of the feature maps may contain unnecessary information for target detection. It is clearly observed that each filter has a different weight that generates a different perspective view of the feature maps. These obtained feature maps are inconsistent for the target objects, which degrades the detection ability from the images. Therefore, grouping the similar feature maps increases the possibility of delivering mostly target-related information to the backbone. In addition, we modified the sizes of the receptive fields of the algorithm for traffic signs in the images. The size of the traffic sign in the image is very small compared to other objects, such as people, cars, and buildings. The sizes of the receptive fields in most algorithms are designed for versatile objects and not traffic signs. Currently, the receptive field size for a large object is not sufficient for traffic signs because traffic signs are much smaller than other objects. Finally, we adopt the ADM-Net [44] as a classifier in the detection algorithm. The ADM-Net comprises the attentional-deconvolution module (ADM) within the network to maximize the feature information with the attention mechanism. The purpose of the ADM-Net here is to specifically classify traffic signs, and it is a suitable classifier for the GTSDB dataset.

The main contributions of this study are as follows:

1.  The inner structure of the backbone involves feature selection to group similar feature maps from the convolution layer using L1-norm. Since the backbone is based on the CSPDarknet [45], the feature maps are divided into two parts. One part of the feature maps is passed on to the next convolution layer for feature extraction, while the other part is concatenated with the transition layer. The L1-norm-based feature selection is applied to the latter concatenated feature maps, and the feature maps with low L1-norm values are removed.

2.  The sizes of the receptive fields of the algorithm are adjusted for traffic signs instead of versatility. The receptive field sizes of most detection algorithms are designed for versatile situations, but, these are not appropriate for traffic signs, owing to their small sizes compared to other objects. Most detection algorithms have three different receptive fields for large-, medium-, and small-sized objects in general. However, the sizes of the receptive fields for small- and medium-sized objects are not designed adequately for traffic signs because the traffic signs are obviously smaller than the other objects.

3.  ADM-Net is adopted as the inference model in the proposed algorithm. ADM-Net is designed to extract the feature information of German traffic signs by considering various external noises. Since the GTSDB dataset is used to detect German traffic signs in the images, the ADM-Net that was specially designed to classify German traffic signs shows superior performance in the experiments.

The remainder of this paper is organized as follows. Section 2 explains the architecture of the proposed FSADD approach. Section 3 discusses the performance comparisons of the FSADD. Finally, Section 4 summarizes and presents the conclusions of this work.

## 2. Network Architecture

This section explains the proposed FSADD architecture in detail. The architecture of the FSADD is based on YOLOv5, along with feature selection, modified receptive fields, and the ADM-Net classifier.

### 2.1. Structure of YOLOv5

Object detection algorithms can be mainly divided into one-stage and two-stage detectors. Two-stage detectors, such as the R-CNN, Fast R-CNN, and Faster R-CNN, usually have high accuracies with slow operation times because of the multiple candidate regions generated. However, one-stage detectors are meant for real-time algorithms with low accuracies and rapid operation times. Therefore, many researchers have studied one-stage detectors to achieve an improved accuracy and speed. Combining the advantages from the previous YOLO versions, YOLOv5 [40], which is an advanced version of YOLOv4 [45], achieves improved accuracy with high speed. Owing to the relatively small structure of YOLOv5, it achieves real-time detection [46].

The structure of YOLOv5 is described in terms of the backbone, neck, and head. The backbone of YOLOv5 is based on the Darknet53 combined with the cross-stage partial network (CSPNet) [47]. The basic purpose of CSPNet is to divide the obtained feature maps from the convolution layer into two parts. The first part is linked to the convolution layer to extract feature information, while the second part is directly passed to the output of the layer. Following the structure of CSPNet, YOLOv5 uses CSPDarknet53 as the backbone to increase the efficiency of the learning ability and remove the computational costs from bottlenecks. The neck of YOLOv5 uses a feature pyramid network (FPN) [48] and path aggregation network (PAN) [49]. The feature maps in the CNN have high-level and low-level features. The high-level features focus on partial or entire objects themselves, while the low-level features contain edge, corner, and color information. Fusing FPN and PAN in YOLOv5 conveys the low- and high-level features through the bottom-up and top-bottom approaches to preserve the strong properties of the features. The head of YOLOv5 includes the bounding box loss function and non-maximum suppression. The receptive fields of YOLOv5 are designed as $80 \times 80$, $40 \times 40$, and $20 \times 20$ for small, medium, and large objects in the image, respectively. With the existence of non-maximum suppression, the redundant bounding boxes with low scores are eliminated as prediction candidates, and the bounding boxes with the highest scores are selected for detection.

### 2.2. L1-Norm Feature Selection

The main approach of the proposed FSADD is to apply feature selection to the backbone, which is the CSPDarknet53. Feature maps obtained from the convolution layer in the CNN are heavily affected by the weights of the filters used during feature extraction and training. The characteristics of the images highlighted by the weights of the filters are different, and each expression of the obtained feature maps is distinct. Some of the obtained feature maps from the convolution layer may provide useful information about the target objects in the image, while some of the feature maps are not related to the target objects. Therefore, determining similarities among the feature maps from the convolution layer is needed in the backbone.

In this work, we adopt the L1-norm in CSPDarknet53 to improve the detection ability. In computer vision, the L1-norm is used to measure feature similarities among images [50,51], prune filters, or feature maps by removing the low values in the CNN [41–43,52]. Based on a different perspective view of the L1-norm usage in the CNN, we apply the L1-norm to the feature selection to group similar feature maps. In general, the L1-norm value of each feature map is different, and feature maps with low values of the L1-norm indicate that their expectations of the magnitude are weak compared to those of other feature maps [41]. A low L1-norm of the feature map results in weak activation inside the CNN, which has a minuscule influence on the detection ability of the network.

However, the network still retains all feature maps from the backbone, which increases computation costs and degrades the operational speed of the detection algorithm.

Figure 1 shows some examples of the original images from the GTSDB dataset and their feature maps sorted by the L1-norm from CSPDarknet53. Figure 1a contains one traffic sign that provides direction for the vehicles on the road, and this needs to be detected. Figure 1b shows the feature maps sorted by L1-norm in order from the lowest to highest. In contrast to the ability of humans' recognition, computer vision algorithms are struggling to identify the structural outlines of the specific objects in the image. Owing to the characteristics of the convolution layer as the feature extraction, the initial outlines of the objects in the image has become one of the major issues [53]. As the image has a large value of the L1-norm, the traffic sign in the image is emphasized with a bright color, so as to ensure a high possibility of providing such information to the next layer. Feature maps with specific low values of the L1-norm do not provide exact outlines of the objects, and these feature maps are less effective for the performance. Therefore, the extracted feature maps are needed to be grouped into two parts.

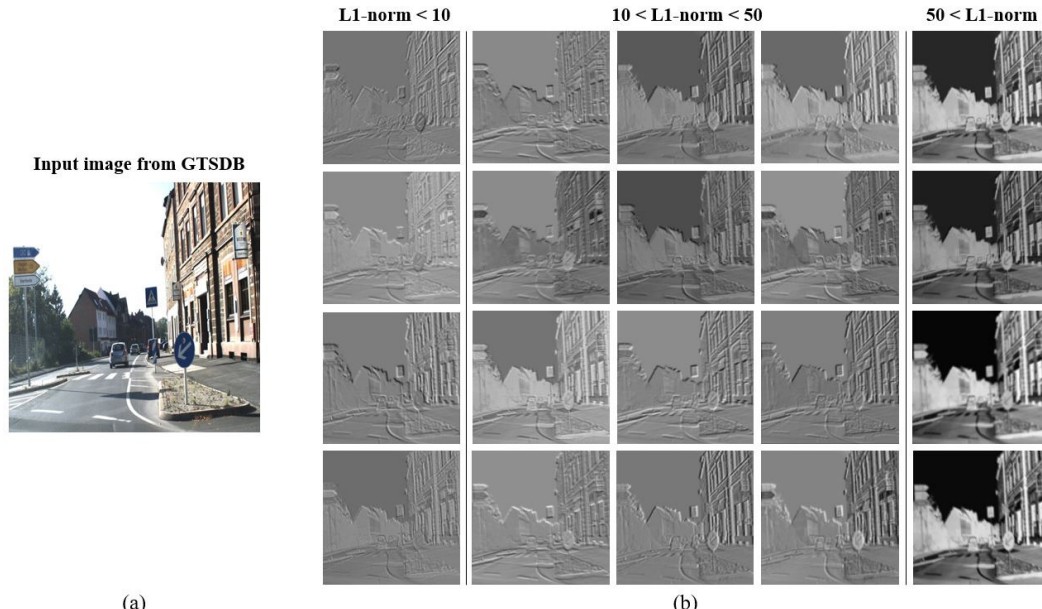

**Figure 1.** Examples of feature maps sorted by L1-norm values: (**a**) original images from GTSDB; (**b**) obtained feature maps from CSPDarknet53.

Figure 2 describes the grouping of similar images based on the L1-norm. Figure 2a is a sample image from the GTSDB dataset with a blue traffic sign located near the center of the image. Figure 2b shows some of the extracted images after the convolution layer. It is observed that the pixel values of the bright regions in the images are highlighted compared with those of the other objects. Grouping images with the highlighted traffic sign may thus improve the traffic sign detection ability. Figure 2c,d show the feature maps from the convolution layer separated by similarity as an example.

Figure 3 shows the complete process of the L1-norm feature selection using the CSP-block from CSPDarknet53. The feature maps are obtained through the convolution layer, batch normalization, and sigmoid linear unit (SiLU). Then, the CSP structure divides the feature maps into two blocks. One of the blocks is conveyed to another convolution layer, followed by batch normalization and SiLU activation to extract feature information. Then, the L1-norm is applied to the other block. The L1-norm value of each feature map from the latter block is computed as follows:

$$\sum_{i=1}^{N} |\mathbf{X}_i| = |x_1| + |x_2| + |x_3| + \cdots + |x_N|. \tag{1}$$

where *i* and *x* are the pixel locations and values from the obtained feature maps. By summing all the pixel values from all pixel locations, the L1-norm value of the feature map is calculated. Then, the L1-norm values of each of the feature maps from the convolution layer are compared to investigate the feature maps that have low effectiveness for detection. Feature maps with low L1-norm values are pruned inside the blocks to reduce the load of the network because these feature maps are less important for detection. Finally, the feature maps retained after pruning are directly linked to the output of the layer.

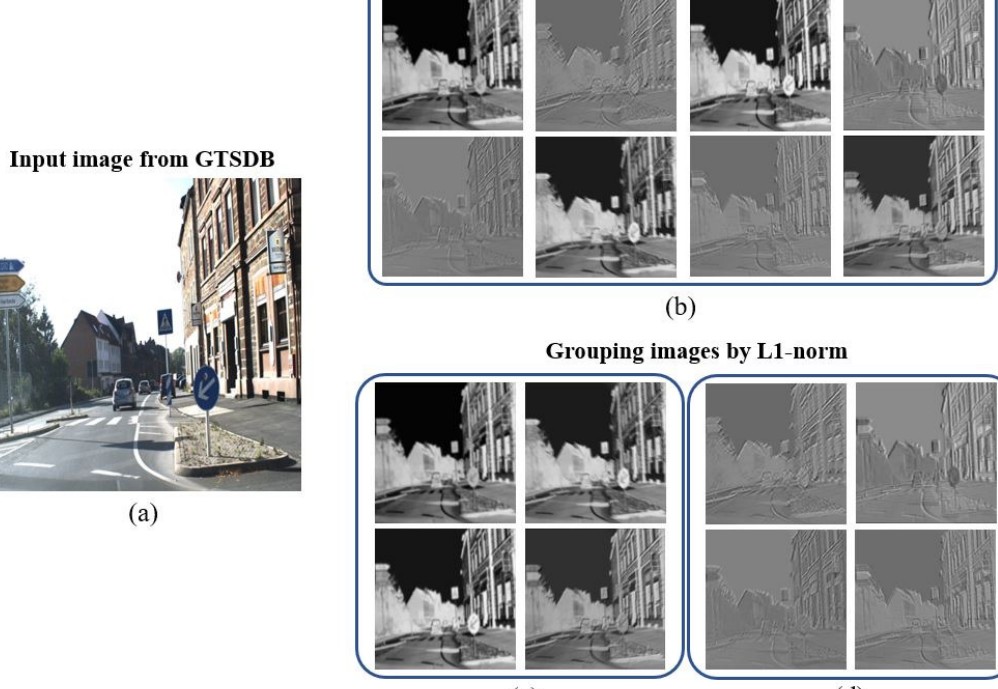

**Figure 2.** Examples of grouping images by similarity: (**a**) original input image from the GTSDB dataset; (**b**) extracted feature maps from the convolution layer; (**c**) grouping feature maps by focusing on the blue traffic sign; (**d**) grouping the remaining feature maps from (**b**).

### 2.3. Feature-Selection-Based Attentional-Deconvolution Detector

The proposed FSADD network is shown in Figure 4. Based on YOLOv5, it consists of the backbone, neck, and head structures. The backbone, which is the CSPDarknet53, has multiple CSP-blocks for feature extraction from the convolution layer, batch normalization, and SiLU activation. The purpose of the proposed network is to detect traffic signs in images, where the major priority is to identify images that highlight traffic signs. Therefore, feature selection based on the L1-norm is applied to each CSP-block to group the highly related similar images from feature extraction. Furthermore, the L1-norm feature selection prunes the low values of each feature map in the network. Feature maps with low L1-norm values imply that these weak activations are less effective for detecting traffic signs. Pruning many feature maps can result in performance degradation; thus, the threshold for the L1-norm value needs to be defined, which is achieved through trial and error. By adjusting the feature selection based on the L1-norm, the network has a high possibility that similar feature maps, which highlight traffic signs based on priority, can be grouped into a single feature block. Thus, the network receives more information about the traffic signs.

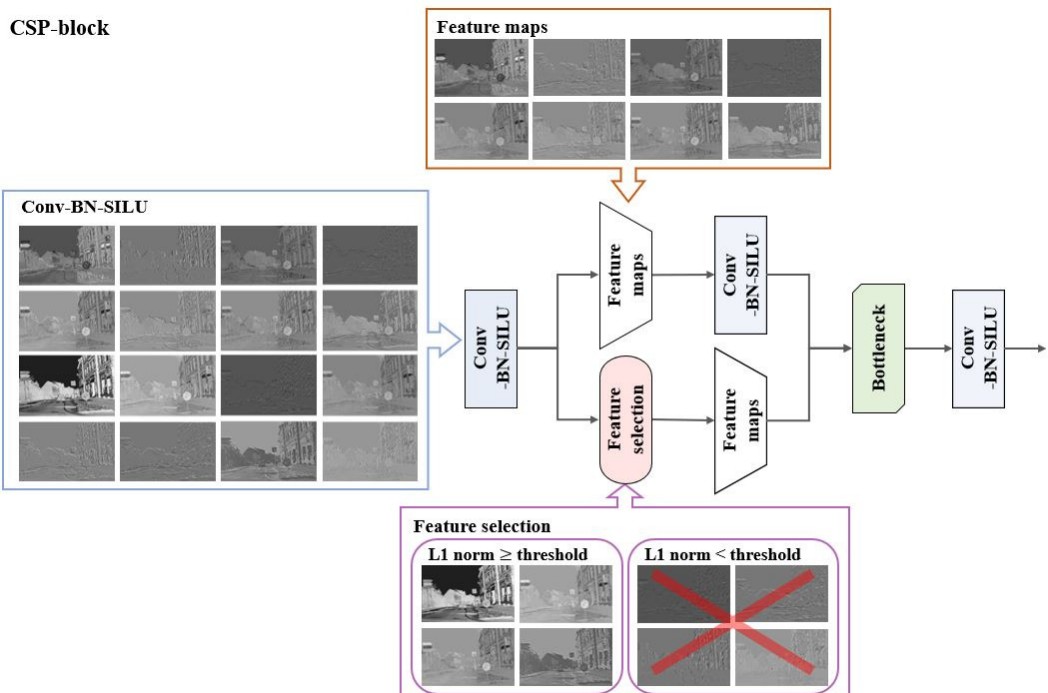

**Figure 3.** Process of the CSP-block in the CSPDarknet53 using L1-norm feature selection.

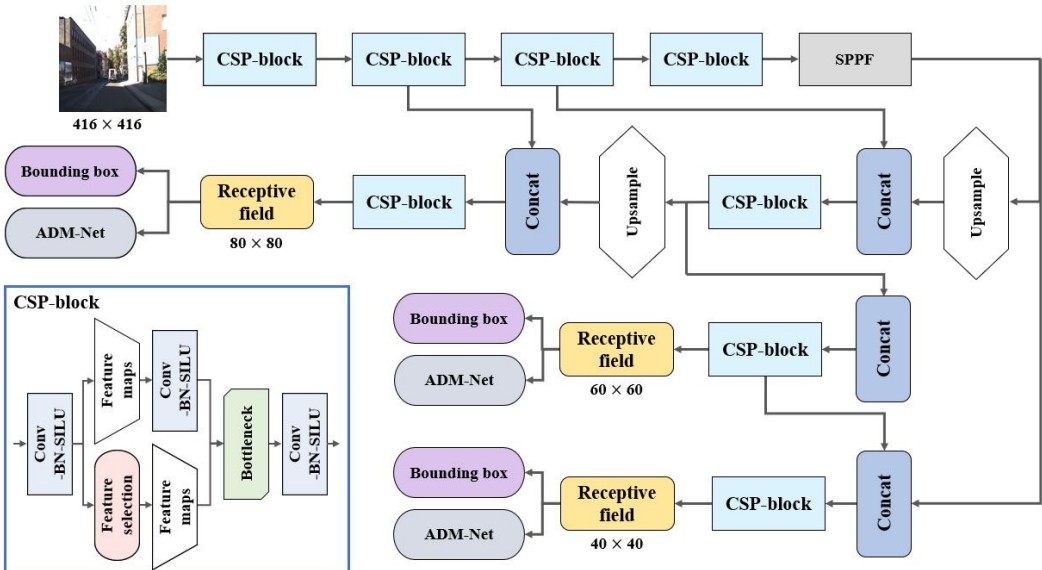

**Figure 4.** Architecture of the proposed feature-selection-based attentional-deconvolution detector (FSADD).

As shown in Figure 4, the sizes of the receptive fields are modified compared with those of the original YOLOv5. The original YOLOv5 has three receptive fields with sizes $80 \times 80$, $40 \times 40$, and $20 \times 20$ for small-, medium-, and large-sized objects. Most detection algorithms have the target of high accuracy with the MS COCO dataset [54], and predesigned receptive fields are not suitable for traffic sign detection. Traffic signs contain limited information to provide the exact status of a road to the driver effectively; thus, the characteristics of the traffic sign must be considered. Road images with traffic signs may include buildings, vehicles, or people, and these objects are often larger than the traffic signs. Sometimes, traffic signs have the smallest size among all objects in the image. The receptive field for a small object, which is predesigned as $80 \times 80$, may not be large enough for detecting traffic signs

in such images. However, doubling the size of the receptive field from $80 \times 80$ increases the computational cost tremendously. Instead of increasing the size of the receptive field for small objects, we modify the receptive fields for medium- and large-sized objects. The traffic sign is made as small as the other objects, so that modifying the sizes of the receptive fields improves the traffic sign detection. Moreover, it is difficult to define the medium- and large-sized traffic signs in the images. Thus, the FSADD has modified receptive fields of sizes $80 \times 80$, $60 \times 60$, and $40 \times 40$ compared with those of the original YOLOv5.

Finally, the proposed FSADD comprises ADM-Net as the traffic sign classifier. A fully connected layer is applied in many CNN-based algorithms for classification owing to its simple and classic structure. For this reason, YOLOv5 adopts a fully connected layer to classify the candidate objects. However, conventional fully connected layers require that the size of the input image be fixed. This property of the fully connected layer limits the ability of the network and does not preserve the spatial coordinate features [55], which is unsuitable for detection. However, ADM-Net solves this drawback of the spatial coordinate features by adding a $1 \times 1$ convolution layer at the end to convert the structure. Moreover, the attention mechanism inside ADM-Net focuses on important regions of the feature maps that enable the network to store highly related information from the images. Because the detection dataset includes multiple objects even in a single image, highlighting the candidate regions tends to be the target object, which improves detection performance. It is commonly known that the details of the objects in images vanish after the convolution layer, and such objects with broken details affect the performance of the network. The deconvolution layer in ADM-Net is adopted to upsample the size of the previously obtained feature map. Feature maps with upsampling not only recover the resolution of the image but also improve the broken details of the objects. Thus, adopting a deconvolution layer helps recover the object shape distortion, and continuous usage of the deconvolution layer minimizes the detail loss of the objects. With the structural advantages of ADM-Net, it is specially designed for the classification of German traffic signs, and this is highly related to the GTSDB dataset. As mentioned, the GTSDB dataset is used to detect 43 kinds of German traffic signs from images. The target detection objects in the GTSDB dataset are part of the German traffic sign classification. Since the main objective of the GTSDB dataset is to detect German traffic signs, replacing the conventional fully connected layer with the ADM-Net, which is specially designed for classifying German traffic signs, is effective owing to the similarities among German traffic signs.

## 3. Traffic Sign Recognition and Comparison Results

### 3.1. Preprocess

The simulation results are conducted with a Core i5-9400F (up to 4.1 GHz), 32 GB DDR4, and GTX 1080Ti. The performance of the FSADD is demonstrated on the GTSDB dataset, which represents various real-time statuses for the 43 kinds of traffic signs. To provide a clear understanding, the 43 kinds of traffic signs from the GTSDB dataset are shown in Figure 5. The numbers in the left most column of Figure 5 show the labels of the traffic signs as ground truths. For example, the speed limit sign with 20 is labeled as 0, while the stop sign is labeled as 14.

Examples from the GTSDB dataset are shown in Figure 6. It is observed that the GTSDB dataset was collected from roads via images of size $1360 \times 800$ pixels. The GTSDB dataset contains 900 images that are not separated into training or test images. Therefore, the training, validation, and test images from the GTSDB need to be divided manually by the users. As shown in Figure 6, some of the images from the GTSDB dataset are stored without traffic signs. In all, about 160 images have missing labels for the ground truths, and 740 images are actual usable images from the GTSDB dataset.

| Labels | 43 types of traffic signs |
|--------|---------------------------|
| 0 ~ 9 | |
| 10 ~ 19 | |
| 20 ~ 29 | |
| 30 ~ 39 | |
| 40 ~ 42 | |

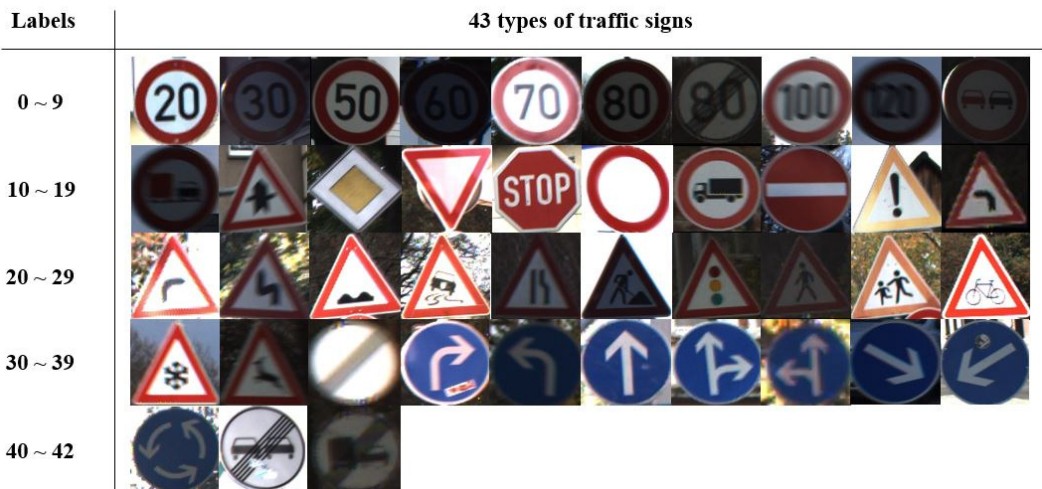

**Figure 5.** Forty-three kinds of German traffic signs from GTSDB dataset.

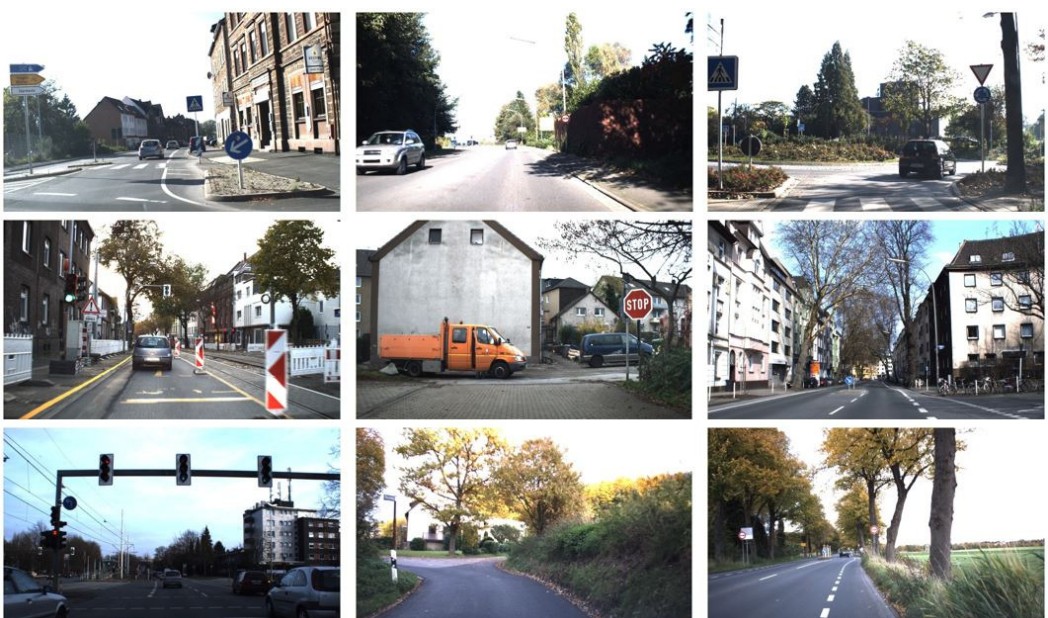

**Figure 6.** Examples from the GTSDB dataset.

Before designing the architecture of the FSADD, the L1-norm threshold for feature selection needs to be defined. Since the FSADD is based on YOLOv5, the L1-norm is applied to the inside of the CSP-block from the original YOLOv5 first to test its effectiveness for different values. During the process, some feature maps with lower L1-norm than the predefined threshold are pruned to increase the detection rate of the traffic signs. Table 1 details the approach for deciding the L1-norm for feature selection. The performance evaluation criteria are defined as follows:

$$\text{Accuracy (\%) of each label} = \frac{\text{\# of correctly detected signs}}{\text{\# of detected signs}} \times 100, \tag{2}$$

$$\text{Total accuracy (\%) of classification} = \frac{\text{Total \# of correctly detected signs}}{\text{Total \# of detected signs}} \times 100. \tag{3}$$

**Table 1.** L1-norm selection for YOLOv5.

| | # of Detected Traffic Sign Types | # of Detected Traffic Signs | # of Correctly Detected Signs | Accuracy (%) |
|---|---|---|---|---|
| L1-norm = 5 | 20 | 158 | 85 | 53.7 |
| L1-norm = 10 | 22 | 162 | 92 | 56.7 |
| L1-norm = 20 | 18 | 164 | 90 | 54.8 |
| L1-norm = 30 | 18 | 140 | 72 | 51.4 |
| YOLOv5 (No L1-norm) | 15 | 123 | 53 | 51.2 |

As shown in Table 1, where # means the number, selecting the L1-norm as 10 results in the detection of more types of traffic signs, with the highest accuracy based on Equation (3). Setting the L1-norm at 10 can recognize 22 out of the 43 types of traffic signs, while the remaining 18 traffic signs are not identified. Setting the L1-norm at 10 proves that not only are higher numbers of traffic sign types detected but also that these have the highest accuracies among other choices. In addition, the number of detected traffic signs is slightly lower when the L1-norm is set to 20. The detection performance of YOLOv5 with GTSDB results in 51.2% accuracy. However, applying the L1-norm as 10 to YOLOv5 achieves 56.7% accuracy, and it detects seven additional kinds of traffic signs. Therefore, the L1-norm for feature selection is set to 10 and applied to each CSP-block.

Table 2 shows the sizes of the receptive fields inside the proposed algorithm that have outstanding performances. It is known that the original YOLOv5 has three different sizes of receptive fields, which are $80 \times 80$, $40 \times 40$, and $20 \times 20$, and results in 51.2% accuracy with the GTSDB dataset. However, the preset for the receptive fields of medium- and large-sized objects is inappropriate for detecting traffic signs because of the specificity of the traffic signs by size. Considering the comparisons in Table 1, modifying the sizes of the receptive fields and setting the L1-norm as 10 is the best approach. By setting the L1-norm to 10, it is demonstrated that changing the sizes of the receptive fields for the medium and large objects results in a more superior performance than others. Receptive fields of sizes $80 \times 80$, $60 \times 60$, and $10 \times 10$ obtain the largest number of detected traffic signs, but the overall performance with $80 \times 80$, $60 \times 60$, and $40 \times 40$ is better. Therefore, the sizes of the receptive fields are set to $80 \times 80$, $60 \times 60$, and $40 \times 40$.

**Table 2.** Selection of receptive field sizes based on the L1-norm.

| | Size of Receptive Fields | # of Detected Traffic Sign Types | # of Detected Traffic Signs | # of Correctly Detected Signs | Accuracy (%) |
|---|---|---|---|---|---|
| L1-norm = 10 | 80, 60, 40 | 28 | 219 | 146 | 66.6 |
| L1-norm = 10 | 80, 60, 10 | 24 | 257 | 144 | 56.0 |
| L1-norm = 10 | 80, 40, 10 | 25 | 198 | 95 | 47.9 |
| YOLOv5 (No L1 norm) | 80, 40, 20 | 15 | 123 | 63 | 51.2 |

Table 3 shows the extra hyper parameter selection for YOLOv5, YOLOv6 [56], YOLOv7, and the FSADD. The weight decay, number of epochs, learning rate, momentum, and optimizer for the detection algorithms are the same.

**Table 3.** Hyper parameter selections for YOLOv5, YOLOv6, YOLOv7, and FSADD.

|        | Weight Decay | Epoch | Learning Rate | Momentum | Optimizer |
|--------|--------------|-------|---------------|----------|-----------|
| YOLOv5 | 0.0005       | 300   | 0.01          | 0.937    | SGD       |
| YOLOv6 | 0.0005       | 300   | 0.01          | 0.937    | SGD       |
| YOLOv7 | 0.0005       | 300   | 0.01          | 0.937    | SGD       |
| FSADD  | 0.0005       | 300   | 0.01          | 0.937    | SGD       |

*3.2. Evaluation and Comparison Results*

3.2.1. Detection Performance of the FSADD

This section demonstrates the performance comparisons of the FSADD and other state-of-the-art detection algorithms using the GTSDB. A total of 740 out of 900 images from the GTSDB are used for the demonstration because the remaining images are missing their ground truths. To demonstrate the detection performance of the FSADD, we divide the GTSDB dataset into training, validation, and test data. The sizes of the images are reduced to $416 \times 416$ instead of the original $1360 \times 800$ because of hardware limitations. Out of the 740 images from the GTSDB dataset, 450 images are set as training, 190 images are set as validation, and 100 images are set as test data.

Table 4 shows the performance of traffic sign recognition of the FSADD using images from the GTSDB. Referring to Figure 5, Label in Table 4 indicates the detected types of traffic signs in the test images. FSADD detects 29 out of the 43 kinds of traffic signs, and a total of 219 traffic signs are recognized. The number of detected traffic signs are divided into two groups as incorrectly and correctly detected traffic signs. The number of incorrectly detected traffic signs contains some traffic signs with wrong labels or other objects that are not recognized as traffic signs. The correctly detected signs in Table 4 are the traffic signs that the FSADD successfully detects with correct labels, and a total of 157 signs are recognized with the correct labels. Following Equations (2) and (3), the individual and total accuracies of the traffic signs are calculated and described in Table 4.

It is noted that Table 4 shows not only the successful detection ratio of GTSDB but also the classification ability of each label by the FSADD. Six kinds of traffic signs, which are the labels 15, 19, 26, 29, 33, and 37, result in 0% accuracy, meaning that the FSADD is unable to detect any of these traffic signs in the GTSDB test images. Except for these six traffic signs, 23 other traffic signs were successfully detected. As the ADM-Net is selected as an inference model, the total accuracy of the FSADD is computed by the following Equation (3), and is 73.9% on the GTSDB test images.

Figures 7 and 8 illustrate some detection results using the FSADD. The detection result of each image is written in the image with distinguishable colors. The upper row of images from Figures 7 and 8, which are (a), (b), and (c), are the original images from the GTSDB dataset, while the lower row of images, (d), (e), and (f), show the detection results. Figure 7a has two traffic signs as ground truths, which are "no entry (17)" and "keep right (38)". Figure 7b has "speed limit 120 (8)" and "no overtaking (trucks) (10)" as the ground truths. Two traffic signs, which are "stop (14)" and "go right or straight (36)" traffic signs, are shown in Figure 7c. Figure 7d–f illustrate the detection results of the FSADD. Figure 7a has two traffic signs with different locations, while the two traffic signs in (b) and (c) are close. Referring to Figure 7, the ground truth values of Figure 8 are "speed limit 50 (2)" and "no overtaking (9)", "speed limit 30 (1)", and "no traffic both ways (15)".

Following the original GTSDB images from Figures 7 and 8, some of the traffic signs are hard to see, owing to the surrounding dark environment, or they contain very tiny portions of the images compared with other objects. Moreover, some of the traffic signs are closely located and need to be recognized separately. Without providing accurate information about the locations of the traffic signs or characteristics during training, the detection algorithm is limited in its ability to locate traffic signs or detect multiple objects at once. This shows that the detection algorithm is unable to detect objects even if the

target object exists or that it detects multiple objects as one object. Therefore, detection algorithms related to traffic signs need to recognize traffic signs as accurately as possible. Furthermore, human eyes are unable to detect traffic signs owing to the surrounding environment, such as the illumination or size of the traffic sign, and providing accurate information about these traffic signs is important. Under such cases, the L1-norm assists in grouping multiple obtained feature maps by similarity and discards those feature maps with weak activations. The continuous usage of feature maps with weak activation is less important during training, and it does not affect the improvement of the detection ability. In addition, the sizes of the traffic signs are generally smaller than other objects; thus, the sizes of the receptive fields need to be modified for detecting traffic signs.

**Table 4.** Traffic sign recognition results of the FSADD for the GTSDB dataset.

| | Label | # of Detected Traffic Signs | # of Incorrectly Detected Signs | # of Correctly Detected Signs | Accuracy (%) |
|---|---|---|---|---|---|
| | 1 | 11 | 3 | 8 | 72.7 |
| | 2 | 6 | 0 | 6 | 100.0 |
| | 3 | 5 | 1 | 4 | 80.0 |
| | 4 | 12 | 1 | 11 | 91.6 |
| | 5 | 7 | 2 | 5 | 71.4 |
| | 7 | 6 | 0 | 6 | 100.0 |
| | 8 | 6 | 1 | 5 | 83.3 |
| | 9 | 8 | 0 | 8 | 100.0 |
| | 10 | 20 | 1 | 19 | 95.0 |
| | 11 | 15 | 0 | 15 | 93.3 |
| | 12 | 16 | 3 | 13 | 81.2 |
| | 13 | 11 | 1 | 10 | 54.5 |
| Types of traffic sign | 14 | 4 | 0 | 4 | 100.0 |
| | 15 | 12 | 12 | 0 | 0.0 |
| | 16 | 2 | 1 | 1 | 50.0 |
| | 17 | 10 | 0 | 10 | 100.0 |
| | 18 | 10 | 4 | 6 | 60.0 |
| | 19 | 3 | 3 | 0 | 0.0 |
| | 20 | 12 | 11 | 1 | 8.3 |
| | 22 | 2 | 0 | 2 | 100.0 |
| | 23 | 3 | 1 | 2 | 66.6 |
| | 25 | 4 | 0 | 4 | 100.0 |
| | 26 | 2 | 2 | 0 | 0.0 |
| | 29 | 1 | 1 | 0 | 0.0 |
| | 32 | 2 | 1 | 1 | 50.0 |
| | 33 | 1 | 1 | 0 | 0.0 |
| | 35 | 9 | 6 | 3 | 33.3 |
| | 37 | 1 | 1 | 0 | 0.0 |
| | 38 | 18 | 0 | 18 | 100.0 |
| Summary | 29 | 219 | | 162 | 73.9 |

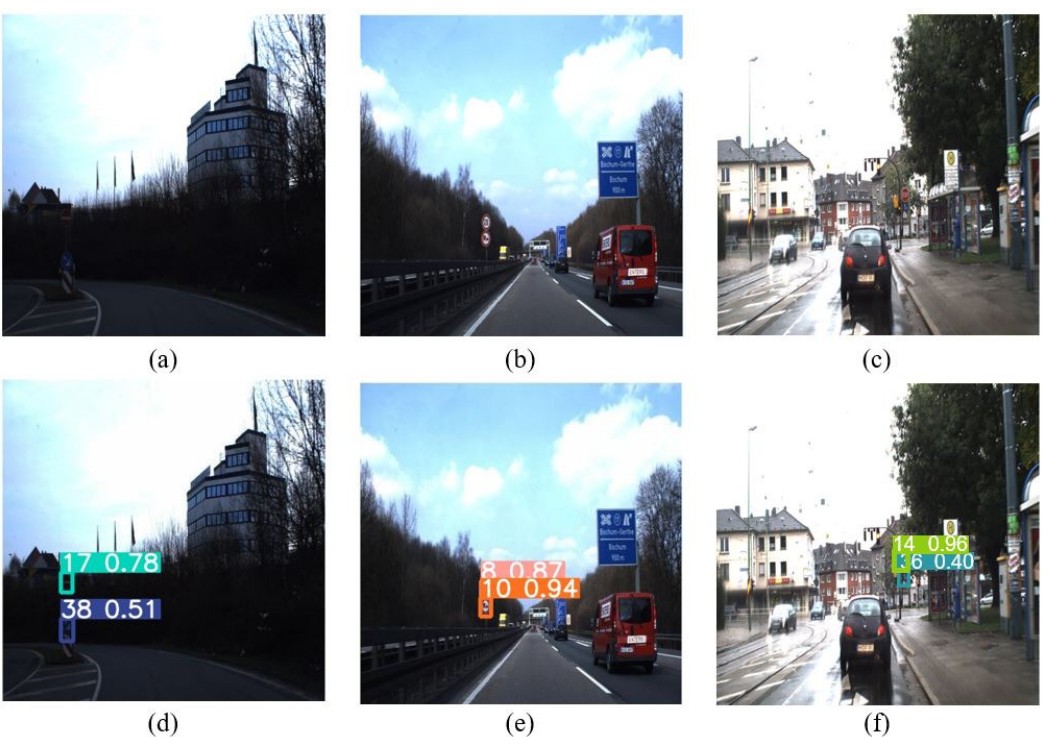

**Figure 7.** Traffic sign detection results using the FSADD—part 1: (**a**) ground truth—17 and 38, (**b**) ground truth—8 and 10, (**c**) ground truth—14 and 36, (**d**) detection results of (**a**), (**e**) detection results of (**b**), and (**f**) detection results of (**c**).

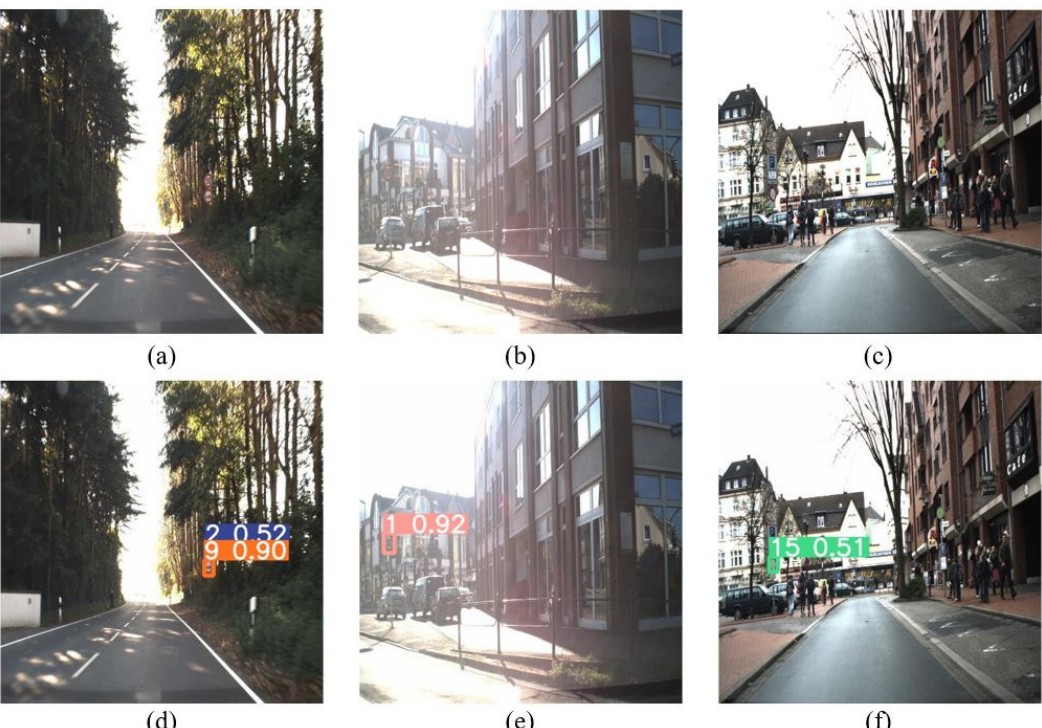

**Figure 8.** Traffic sign detection results using the FSADD—part 2: (**a**) ground truth—2 and 9, (**b**) ground truth—1, (**c**) ground truth—15, (**d**) detection results of (**a**), (**e**) detection results of (**b**), and (**f**) detection results of (**c**).

### 3.2.2. Traffic Sign Recognition Performance Comparisons

Tables 5–7 show the traffic sign recognitions of different state-of-the-art algorithms, which are YOLOv5, YOLOv6, and YOLOv7. Since the newest YOLOv7 contains structural advantages from the previous YOLO series, the inner architecture has become complex with increased parameter numbers. The biggest change between YOLOv5 and YOLOv6 is an efficient decoupled head instead of the conventional YOLO head structure. Compared with traditional YOLO, the head of YOLOv6 has a decoupled structure that separates feature maps from the final head by applying additional layers to increase performance. Therefore, the detection performance of YOLOv6 is superior to YOLOv5 owing to the difference in the structural change of the head. Compared with YOLOv6, YOLOv7 sets bag-of-freebies to perform training optimization without increasing the inference cost. Moreover, YOLOv7 suggests a compound model scaling method to take advantage of E-ELAN without its defects so that the network is unable to analyze the scaling factor when the number of channels is changed. As shown in Table 5 through Table 7, YOLOv5 and YOLOv6 detect 15 and 22 kinds of traffic signs. Both YOLOv5 and YOLOv6 detect 123 traffic signs from the GTSDB test images.

**Table 5.** Traffic sign recognition results of YOLOv5 using GTSDB.

|  | Label | # of Detected Traffic Signs | # of Incorrectly Detected Signs | # of Correctly Detected Signs | Accuracy |
|---|---|---|---|---|---|
| | 1 | 13 | 9 | 4 | 30.7 |
| | 2 | 19 | 12 | 7 | 36.8 |
| | 5 | 13 | 9 | 4 | 30.7 |
| | 6 | 5 | 5 | 0 | 0.0 |
| | 7 | 7 | 4 | 3 | 42.8 |
| | 8 | 8 | 3 | 5 | 62.5 |
| Types of traffic sign | 9 | 2 | 1 | 1 | 50.0 |
| | 10 | 11 | 2 | 9 | 81.8 |
| | 12 | 12 | 5 | 7 | 58.3 |
| | 13 | 5 | 1 | 4 | 80.0 |
| | 14 | 4 | 2 | 2 | 50.0 |
| | 17 | 6 | 1 | 5 | 83.3 |
| | 25 | 1 | 1 | 0 | 0.0 |
| | 30 | 3 | 3 | 0 | 0.0 |
| | 38 | 14 | 0 | 12 | 85.7 |
| Summary | 15 types | 123 | | 63 | 51.2 |

The accuracy of each label with respect to correctly detected traffic signs is computed using Equation (2). YOLOv5 recognizes some of the traffic signs as label 6, 25, and 30, but the traffic signs with respect to these labels are incorrectly detected. Traffic signs with label 38 have the highest accuracy compared to other traffic signs. YOLOv5 achieves 51.2% accuracy from Equation (3). Compared to YOLOv5, YOLOv6 detects eight more kinds of traffic signs. While YOLOv5 is unable to detect three traffic signs that have labels 6, 25, and 30, YOLOv6 recognizes label 25. In contrast, YOLOv6 can detect labels 6 and 25 but not 30. However, YOLOv5 and YOLOv6 have 0% accuracy for label 6, which means that both algorithms cannot identify label 6. YOLOv6 achieves a total 67.4% accuracy, which is higher than that of YOLOv5. Compared to YOLOv5 and YOLOv6, YOLOv7 has inferior performance on the GTSDB dataset. YOLOv7 identifies 13 types of traffic signs, and 72 traffic signs are counted from the test images. The performance of YOLOv7 is demonstrated on the MS COCO dataset, where it records the highest performance compared with other state-of-the-art algorithms. However, the performance of YOLOv7

with the GTSDB dataset shows different aspects than that for the MS COCO dataset. Thus, applying YOLOv7 to the GTSDB dataset must be studied further in the future.

**Table 6.** Traffic sign recognition results of YOLOv6 using GTSDB.

| | Label | # of Detected Traffic Signs | # of Incorrectly Detected Signs | # of Correctly Detected Signs | Accuracy (%) |
|---|---|---|---|---|---|
| | 1 | 3 | 0 | 3 | 100.0 |
| | 2 | 12 | 8 | 4 | 33.3 |
| | 4 | 6 | 3 | 3 | 50.0 |
| | 5 | 11 | 8 | 3 | 27.2 |
| | 6 | 2 | 2 | 0 | 0.0 |
| | 7 | 9 | 5 | 4 | 44.4 |
| | 8 | 6 | 2 | 4 | 66.6 |
| | 9 | 8 | 2 | 6 | 75.0 |
| | 10 | 11 | 2 | 9 | 81.8 |
| | 11 | 6 | 0 | 6 | 100.0 |
| Types of traffic sign | 12 | 9 | 0 | 9 | 100.0 |
| | 13 | 4 | 0 | 4 | 100.0 |
| | 14 | 2 | 0 | 2 | 100.0 |
| | 16 | 1 | 0 | 1 | 100.0 |
| | 17 | 6 | 0 | 6 | 100.0 |
| | 18 | 5 | 3 | 2 | 40.0 |
| | 20 | 1 | 1 | 0 | 0.0 |
| | 23 | 1 | 1 | 0 | 0.0 |
| | 25 | 2 | 0 | 2 | 100.0 |
| | 26 | 1 | 1 | 0 | 0.0 |
| | 36 | 2 | 1 | 1 | 50.0 |
| | 38 | 15 | 1 | 14 | 93.3 |
| Summary | 22 types | 123 | | 83 | 67.4 |

**Table 7.** Traffic sign recognition results of YOLOv7 using GTSDB.

| | Label | # of Detected Traffic Signs | # of Incorrectly Detected Signs | # of Correctly Detected Signs | Accuracy (%) |
|---|---|---|---|---|---|
| | 1 | 18 | 16 | 2 | 11.1 |
| | 2 | 16 | 13 | 3 | 18.7 |
| | 7 | 3 | 3 | 0 | 0.0 |
| | 8 | 3 | 2 | 1 | 33.3 |
| | 10 | 4 | 1 | 3 | 75.0 |
| | 11 | 2 | 1 | 1 | 50.0 |
| Types of traffic sign | 12 | 2 | 0 | 2 | 100.0 |
| | 13 | 2 | 1 | 1 | 50.0 |
| | 14 | 5 | 3 | 2 | 40.0 |
| | 25 | 1 | 0 | 1 | 100.0 |
| | 30 | 2 | 2 | 0 | 0.0 |
| | 33 | 1 | 1 | 0 | 0.0 |
| | 38 | 13 | 3 | 10 | 76.9 |
| Summary | 13 types | 72 | | 26 | 36.1 |

Figure 9 illustrates the detection results of YOLOv5, YOLOv6, YOLOv7, and the FSADD. Ground truths of these images are explained in Figures 7 and 8. As shown in Figure 9a, two types of traffic signs need to be detected by the detection algorithms. YOLOv5, YOLOv6, and the FSADD detect two types of traffic signs, while YOLOv7 detects only one traffic sign. Ground truth values of Figure 9a are defined as labels 2 and 9, where YOLOv5 and the FSADD identify these two signs correctly. From Figure 9b, "speed limit 30", which is labeled as 1, exists in the image. Excluding YOLOv7, each detection algorithm recognizes that the traffic sign is label 1, but the FSADD has the highest score among the YOLOs. Label 15, which is defined as "no traffic both ways", is shown in Figure 9c. YOLOv5 judges that it is label 12, whose ground truth is "priority road", but, it is incorrectly detected. Ground truth values of Figure 9d are labels 17 and 38. In this case, all detection algorithms successfully identify the traffic sign label 38. However, YOLOv6 and the FSADD detect label 17 in the image. In contrast, YOLOv6 has a high score for label 38, while label 17 from the FSADD is higher. Two traffic signs as labels 8 and 10 are recognized by all detection algorithms in Figure 9e. However, the scores of YOLOv7 record the lowest values among the algorithms. YOLOv6 shows the high values of the two traffic signs equally, versus YOLOv5, but the FSADD achieves a higher score than YOLOv6.

YOLOv6 and the FSADD detect the two traffic signs, while YOLOv5 and YOLOv7 detect only one traffic sign in Figure 9f. This shows that the performances of YOLOv6 and the FSADD in Figure 9f are quite similar.

Figure 10 is another detection performance comparison of YOLOv5, YOLOv6, YOLOv7, and the FSADD. Ground truth values of Figure 10a are labeled as 7, where the FSADD identifies the signs more correctly than others. Figure 10b has one traffic sign, which says "no overtaking", and is defined as label 9.

YOLOv5, YOLOv6, and YOLOv7 predict the location of the traffic sign successfully, but those algorithms are unable to identify the traffic sign, excluding the FSADD. YOLOv6 and the FSADD only recognize the traffic sign, which says "priority at next intersection", but the FSADD has a high score in Figure 10c. From Figure 10d, "no overtaking", which is labeled as 9, is in the image.

YOLOv5 and YOLOv7 show none or wrong results with respect to the traffic sign. YOLOv6 labels the traffic sign correctly, but the FSADD has the highest score in this case. Figure 10e has three types of traffic signs, which are labeled as two "no entry" (17) and one "keep right" (38). Excluding YOLOv7, YOLOv5, YOLOv6, and the FSADD identify three traffic signs equally, but the FSADD scores the highest value. Ground truth values of Figure 10f, which are two signs saying "speed limit 80", are labeled as 5. YOLOv5 detects two traffic signs as label 2, and YOLOv7 judges that there is one traffic sign, which is label 1. YOLOv6 and the FSADD equally recognize two traffic signs as label 5, but the FSADD achieves a higher score than YOLOv6.

Thus, the FSADD shows advantages for detecting traffic signs in various cases. Some of the traffic signs are hard to recognize owing to small sizes or the surrounding environments. The architecture of the FSADD is designed to detect traffic signs effectively regardless of other objects in the images. Applying feature selection with L1-norm groups the similar images together. It allows a high possibility that feature information delivered by similar images includes target objects in the images. In addition, designing receptive fields for specific targets recognizes more kinds of traffic signs instead of using the original sizes of the receptive fields. Replacing the conventional classifier with ADM-Net, which is specialized for traffic signs, achieves high scores of classification compared with other state-of-the-art algorithms.

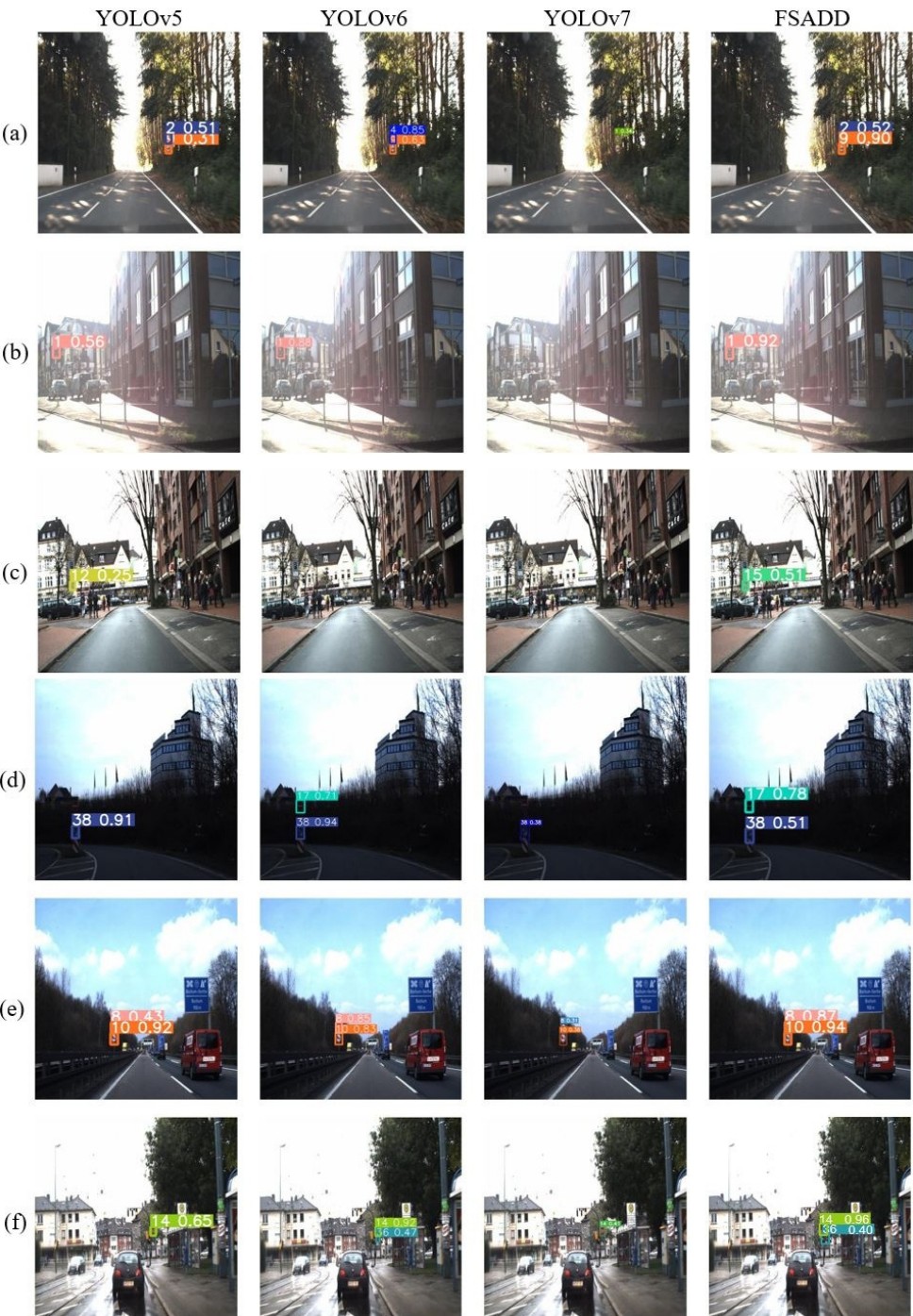

**Figure 9.** Traffic sign detection performance comparisons (YOLOv5, YOLOv6, YOLOv7, and FSADD)—Part 1: (**a**) ground truth—17 and 38, (**b**) ground truth—1, (**c**) ground truth—15, (**d**) ground truth—17 and 38, (**e**) ground truth—8 and 10, and (**f**) ground truth—14 and 36.

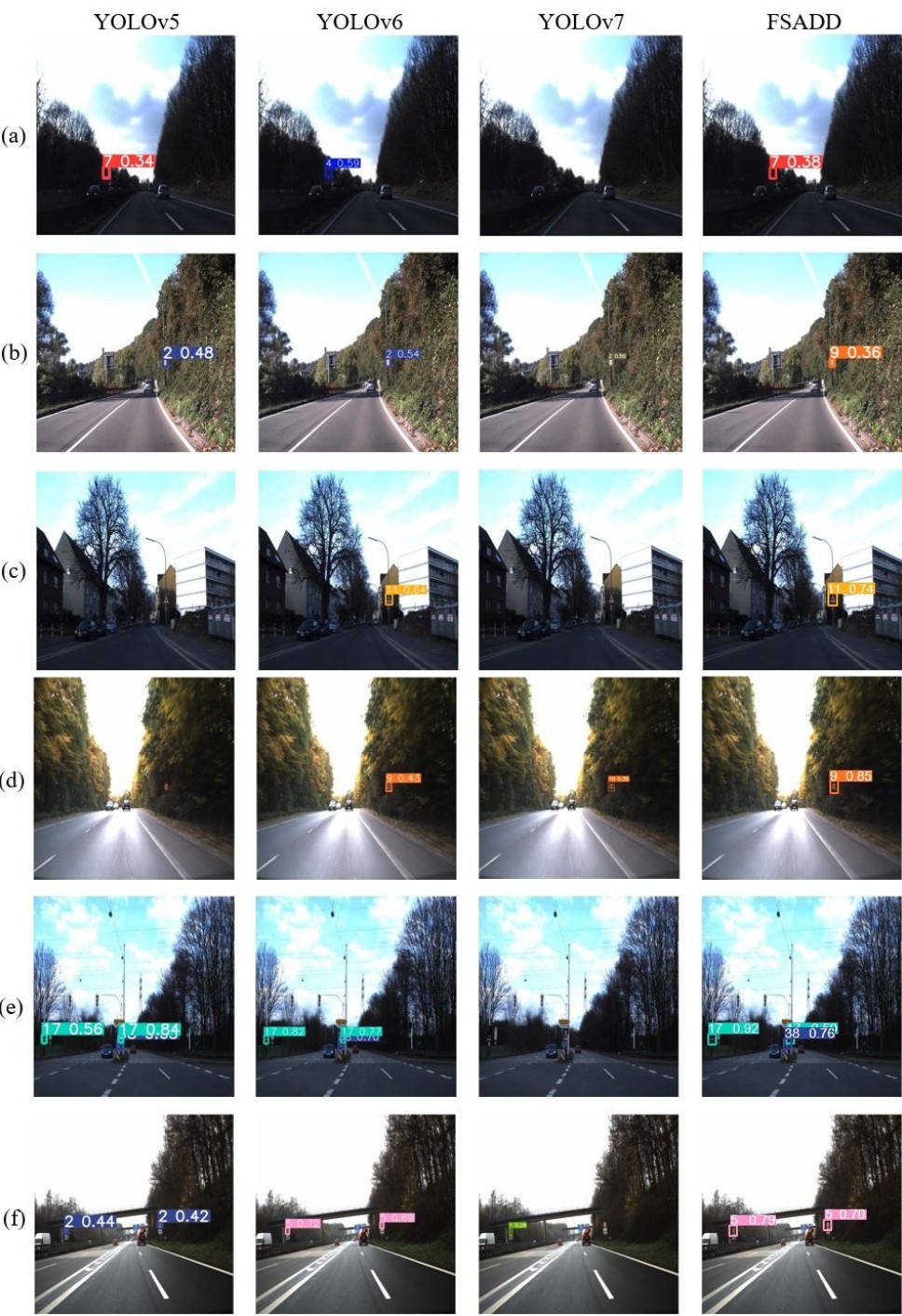

**Figure 10.** Traffic sign detection performance comparisons (YOLOv5, YOLOv6, YOLOv7, and FSADD)—Part 2: (**a**) ground truth—7, (**b**) ground truth—9, (**c**) ground truth—11, (**d**) ground truth—9, (**e**) ground truth—17 and 38, and (**f**) ground truth—5.

## 4. Discussion

Table 8 represents the overall performance comparisons among detection algorithms with the GTSDB dataset. By reducing the sizes of the input image, the size and the resolution of the traffic sign reduce drastically compared to the original size. Searching small objects with low resolution in the image or training with a small size of the training dataset has been challenging in studies with deep-learning-based approaches.

With the reduced size of the GTSDB dataset for training, neither the number of the detected traffic signs nor the accuracy of the Faster R-CNN is recorded when the dataset

size is set to 416 × 416. Many studies have reported that the Faster R-CNN has some limitations in terms of searching for small objects [57,58], losing semantic information due to low resolution [59,60], or dropping performance with the small size of the training dataset [61]. The amount of viable training of the GTSDB dataset is 740 images, and the width and height of the training images are reduced in the implementations. Due to such conditions, it is analyzed that the Faster R-CNN is an invalid method. YOLOv6 detects 22 kinds of traffic signs with 67.4% accuracy, while YOLOv5 achieves 51.2% accuracy with 15 kinds of traffic signs. However, the giga floating-point operations (GFLOPs) of YOLOv6 are approximately three times larger than YOLOv5. The performance of YOLOv7 with the GTSDB dataset shows questionable results. It is demonstrated that the performance of YOLOv7 with the MS COCO dataset shows outstanding results versus other state-of-the-art algorithms. The traffic sign detection performance of YOLOv7 with the GTSDB dataset needs to be analyzed further in the future.

**Table 8.** Traffic sign recognition comparisons using GTSDB.

|  | Dataset Size | Types of Detected Traffic Signs | # of Detected Traffic Signs | Accuracy (%) | GFLOPs |
|---|---|---|---|---|---|
| Faster R-CNN -FPN [32,62] | 1360 × 800 | 35 | 168 | 91.6 | 180 |
| Faster R-CNN -FPN [32,62] | 416 × 416 | - | - | - | 180 |
| YOLOv5 | 416 × 416 | 15 | 123 | 51.2 | 16.2 |
| YOLOv6 | 416 × 416 | 22 | 123 | 67.4 | 44 |
| YOLOv7 | 416 × 416 | 13 | 72 | 36.1 | 104.5 |
| FSADD (Proposed) | 416 × 416 | 29 | 219 | 73.9 | 16.63 |

The proposed method, FSADD, shows the largest number of detected traffic signs. Compared to the increase in accuracy, the increase in the FSADD's GFLOPs is insignificant. Designing the FSADD with a feature selection based on the L1-norm, modifying the sizes of the receptive fields, and adding ADM-Net as the inference model, produces an outstanding performance on the GTSDB dataset. The highest accuracy of the FSADD means that it identifies more traffic signs correctly even if the FSADD detects a higher number of traffic signs.

Figure 11 shows the error cases of the traffic sign recognition by the FSADD. Ground truth values of Figure 11a are labeled as 1 and 25 but the different object in the image is identified. Figure 11b has one traffic sign, which is "speed limit 50" (2) as the ground truth; however, the FSADD recognizes it as label 3. FSADD detects a traffic sign and labels it as "speed limit 60" (3) but the ground truth value of Figure 11c is "speed limit 100" (7). Figure 11d,e have one traffic sign but the ground truth values of each image are "speed limit 80" (5) and "danger" (18). From Figure 11f, "keep right" (38) is the ground truth value but the FSADD defined the traffic sign as label 15.

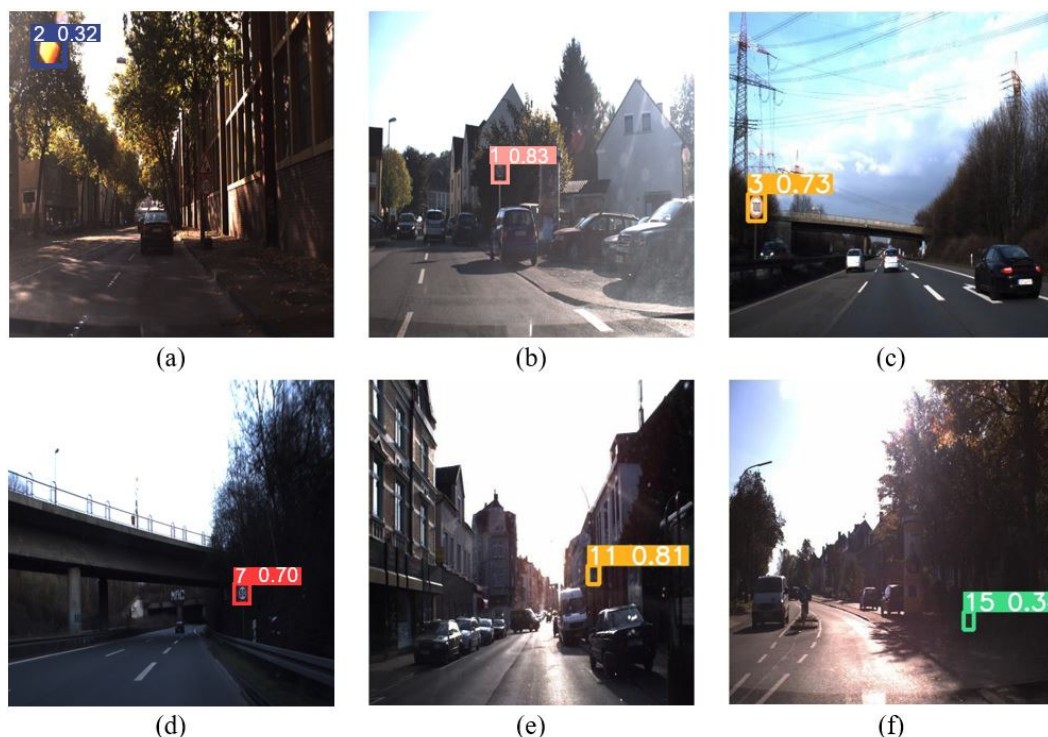

**Figure 11.** False cases of traffic sign recognition by FSADD: (**a**) ground truth—1 and 25, (**b**) ground truth—2, (**c**) ground truth—7, (**d**) ground truth—5, (**e**) ground truth—18, and (**f**) ground truth—38.

## 5. Conclusions

The FSADD, a novel detection algorithm for traffic signs, is proposed in this paper. The structure of the FSADD consists of a feature selection with L1-norm for the backbone, modified sizes of receptive fields, and ADM-Net as the inference model. Applying L1-norm to the obtained feature maps inside CSP-block, it groups feature maps by similarity and prunes some of the feature maps with weak activation. Owing to this strategy, the backbone of the FSADD tends to utilize feature maps with strong activations to improve the usages of the feature map information. Adjusting the sizes of receptive fields in the FSADD for the GTSDB dataset leads to improvement in terms of detecting traffic signs from candidate regions. The size of traffic signs are mostly smaller than other objects in the images, and pre-defined sizes of receptive fields are not ideal. Most state-of-the-art detection algorithms consider high generality in real applications and that their structures are not empirical for detecting traffic signs. Combining ADM-Net as an inference model in the FSADD exhibits the highest accuracy.

Detection performances of the FSADD are demonstrated through simulation evaluations, and the FSADD outperforms other state-of-the-art algorithms. Traffic signs have distinct characteristics versus other objects; the FSADD detects more types of traffic signs accurately. The proposed FSADD detects 29 out of 43 traffic signs, and it achieves 73.9% accuracy using the GTSDB dataset. However, the FSADD is still unable to detect the rest of the traffic signs in the GTSDB dataset and provides 0% accuracy for some specific kinds of traffic signs. Therefore, an improved version of the FSADD will be investigated to increase its detection ability further by designing a modified backbone structure and expanding the structure of the FSADD for different traffic sign datasets in the future.

**Author Contributions:** Conceptualization, J.C., S.P., D.P. and M.L.; methodology, J.C., D.P. and M.L.; software, J.C.; validation, J.C., D.P. and M.L.; formal analysis, J.C.; investigation, J.C. and M.L.; resources, J.C.; data curation, J.C. D.P. and H.C.; writing—original draft preparation, J.C.; writing—review and editing, D.P. and M.L.; visualization, J.C.; supervision, M.L.; project administration, M.L.; funding acquisition, all authors. All authors have read and agreed to the published version of the manuscript.

**Funding:** This research was supported by the Basic Science Research Program through the National Research Foundation of Korea(NRF) (grant no. NRF-2022R1F1A1073543).

**Institutional Review Board Statement:** Not applicable.

**Data Availability Statement:** We used a dataset with figures (https://benchmark.ini.rub.de/gtsdb_news.html, accessed on 1 January 2021).

**Conflicts of Interest:** The authors declare no conflict of interest.

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
