# Peer review of "Feature-Selection-Based Attentional-Deconvolution Detector for German Traffic Sign Detection Benchmark"

_electronics, doi:10.3390/electronics12030725_

Round 1
Reviewer 1 Report
In this paper, the authors propose a propose a novel traffic sign detection algorithm based on the deep learning approach and perform experiments evaluating the proposed model. The following defects should be fixed, especially the innovation issue.
1. The developed system is lack of real technical contribution as the employed methods are all existing techniques. So this paper does not tell people new advancements.
2. The authors should provide more motivations of the proposed solution. A typical example is Fig.1 and Fig. 2. The authors focus too much on what it is, but ignores where it comes from and why it is.
3. The authors should clarify the experimental environment and experimental settings. As the authors aim to develop a novel traffic sign detection algorithm, the characteristics of an system should be given in the experiment section.
4. The authors claimed in the result section that "The highest accuracy of the FSADD means that it identifies more traffic signs correctly even is FSADD detects more number of traffic signs. " but the reason for the same is not reflected from the results?
5. In terms of evaluation indicators, in addition to detection and effective classification performances, it should also include the comparison results of error indicators.
6. In the Experiment Results part, the experimental study is rather incomplete and needs to be further enriched. Adding more proof data and experimental details is necessary.
7. Overall, the manuscript's structure still needs to be updated since, at some points, the manuscript talks about one thing and then indirectly contradicts the same thing at other points.
8. I am afraid the English writing is hard to understand. The presentation and language use of this paper should be further improved.
Reviewer 2 Report
Dear authors,
Paper is good. Please do the following changes for the acceptance of the manuscript.
1. Mention best results obtained in abstract
2. Yolo and CNN architectures have to be reviewed. Please include these related papers.
a. Krishnadas P, Chadaga K, Sampathila N, Rao S, Prabhu S. Classification of Malaria Using Object Detection Models. InInformatics 2022 Sep 27 (Vol. 9, No. 4, p. 76). MDPI.
b.Sampathila N, Chadaga K, Goswami N, Chadaga RP, Pandya M, Prabhu S, Bairy MG, Katta SS, Bhat D, Upadya SP. Customized Deep Learning Classifier for Detection of Acute Lymphoblastic Leukemia Using Blood Smear Images. InHealthcare 2022 Sep 20 (Vol. 10, No. 10, p. 1812). MDPI.
c. Acharya V, Dhiman G, Prakasha K, Bahadur P, Choraria A, Prabhu S, Chadaga K, Viriyasitavat W, Kautish S. AI-assisted tuberculosis detection and classification from chest X-rays using a deep learning normalization-free network model. Computational Intelligence and Neuroscience. 2022 Oct 3;2022.
Other papers also have to be reviewed (another two or three)
3. Add threat to validation section
4. Need to add discussion section and explain the results. Also, compare your work with related literature using a table.
5. Add challenges and future directions
Reviewer 3 Report
In this paper (entitled "Feature-Selection-based Attentional-Deconvolution Detector for German Traffic Sign Detection Benchmark"), authors propose a novel traffic sign detection algorithm based on the deep-learning approach. However, there are some issues that should be addressed before this work can be accepted. The detailed comments are given as follows:
1. Is the algorithm proposed by the author for identifying more types of traffic signs or better performance?
2. It is suggested that the author clearly state what challenges in traffic sign recognition are mainly solved in this paper.
3. Where does the paper show "Attentional-Resolution"?
4. It is suggested that the author analyze the performance differences between FSADD algorithm and other advanced algorithms in the experimental part.
5. The reference list can be enhanced. Some intelligent methods have potential to deal with the problem.
[1] "Evaluation of deep neural networks for traffic sign detection systems." Neurocomputing 316 (2018): 332-344.
[2]"Enhancing Learning Efficiency of Brain Storm Optimization via Orthogonal Learning Design," IEEE Transactions on Systems, Man, and Cybernetics: Systems, vol. 51, no. 11, pp. 6723-6742, Nov. 2021.
[3] " Proceedings of the 11th International Conference on Computer Engineering and Networks. Springer, Singapore, 2022.
The above three references should be included in the revision.
Round 2
Reviewer 1 Report
The revised manuscript can be accepted for publication
Author Response
Thank you for your valuable comments.
Reviewer 3 Report
The presentation of the paper can be improved further. The reference [4] is wrong (the title of the paper is not correct).
Author Response
Thank you for your valuable comments.
The title of the paper in reference [4] is correct.
Round 3
Reviewer 3 Report
No further comments.